# Bioinformatic Analysis for Influential Core Gene Identification and Prognostic Significance in Advanced Serous Ovarian Carcinoma

**DOI:** 10.3390/medicina57090933

**Published:** 2021-09-04

**Authors:** Changho Song, Kyoung-Bo Kim, Jae-Ho Lee, Shin Kim

**Affiliations:** 1Department of Obstetrics and Gynecology, Keimyung University, 1095 Dalgubeol-daero, Dalseo-gu, Daegu 42601, Korea; songchanghomd@gmail.com; 2Department of Laboratory Medicine, Keimyung University, 1095 Dalgubeol-daero, Dalseo-gu, Daegu 42601, Korea; kimbo707@dsmc.or.kr; 3Department of Anatomy, School of Medicine, Keimyung University, 1095 Dalgubeol-daero, Dalseo-gu, Daegu 42601, Korea; anato82@dsmc.or.kr; 4Department of Immunology, School of Medicine, Keimyung University, 1095 Dalgubeol-daero, Dalseo-gu, Daegu 42601, Korea; 5Institute of Medical Science & Institute of Cancer Research, Keimyung University, 1095 Dalguceol-daero, Dalseo-gu, Daegu 42601, Korea

**Keywords:** serous ovarian carcinoma, prognostic biomarker, *CDCA3*, *UBE2C*

## Abstract

*Background and objectives:* Ovarian cancer is one of the leading causes of death among women worldwide. Most newly diagnosed ovarian cancer patients are diagnosed in advanced stages of the disease. Despite various treatments, most patients with advanced stage ovarian cancer, including serous ovarian cancer (the most common subtype of ovarian cancer), experience recurrence, which is associated with extremely poor prognoses. In the present study, we aimed to identify core genes involved in ovarian cancer and their associated molecular mechanisms, as well as to investigate related clinicopathological implications in ovarian cancer. *Materials and methods:* Three gene expression cohorts (GSE14407, GSE36668, and GSE38666) were obtained from the Gene Expression Omnibus databases to explore potential therapeutic biomarkers for ovarian cancer. Nine up-regulated and six down-regulated genes were screened. Three publicly available gene expression datasets (GSE14407, GSE36668, and GSE38666) were analyzed. *Results:* A total of 14 differently expressed genes (DEGs) were identified, among which nine genes were upregulated (*BIRC5*, *CDCA3*, *CENPF*, *KIF4A*, *NCAPG*, *RRM2*, *UBE2C*, *VEGFA*, and *NR2F6*) and were found to be significantly enriched in cell cycle regulation by gene ontology analysis. Further protein–protein interaction network analysis revealed seven hub genes among these DEGs. Moreover, Kaplan–Meier survival analysis showed that a higher expression of *CDCA3* and *UBE2C* was associated with poor overall patient survival regardless of tumor stage and a higher tumor histologic grade. *Conclusion*: Altogether, our study suggests that *CDCA3* and *UBE2C* may be valuable biomarkers for predicting the outcome of patients with advanced serous ovarian cancer.

## 1. Introduction

Ovarian cancer is the eighth cause of cancerous incidence and death in women worldwide, accounting for 3.4% of all of cancer in females and for 4.7% of female cancer deaths due to a low survival rate [1]. Most ovarian cancers are asymptomatic, which contributes to approximately 70% of cases being diagnosed at an already advanced stage that, in turn, will lead to a poor treatment outcome [2,3]. Recent advances in treatment strategies have improved the outcome of patients with ovarian cancer; nonetheless, serous ovarian cancer, which is the most common subtype of ovarian cancer, still has a poor prognosis [4,5]. The remarkable development of bioinformatics has recently enabled the scientific community to research improving treatment outcome and survival of ovarian cancer patients through identifying potential therapeutic targets and prognostic markers [6,7,8]. However, there are no established genetic biomarkers other than the *BRCA1/2* mutation to predict the prognosis of patients with ovarian cancer [9,10]. This highlights the need for a better understanding of the underlying features of ovarian cancer to pave the way for the identification of specific therapeutic targets and prognostic markers, as well as for the development of enhanced treatment strategies.

Recently, very powerful publicly available gene expression databases such as Gene Expression Omnibus (GEO) [11] and The Cancer Genome Atlas (TCGA) cohorts [12] have been widely used. In the present study, we investigated the dysregulated genes and their prognostic significance in serous ovarian carcinoma using GEO datasets and a TCGA Ovarian Cancer (TCGA OV) cohort. First, we identified differentially expressed genes (DEGs) from three GEO datasets by applying the GEO2R online tool and Venn diagram software. Second, Gene Ontology (GO) functional enrichment analysis was conducted with the Database for Annotation, Visualization, and Integrated Discovery (DAVID), including biological process (BP), cellular component (CC), and molecular function (MF), using the identified DEGs. Third, a protein–protein interaction (PPI) network of DEGs was constructed to identify important core genes related to serous ovarian cancer by the Search Tool for the Retrieval of Interacting Genes (STRING) and Cytoscape. Finally, Kaplan–Meier analysis of the selected core genes was performed to evaluate the prognostic potential of the DEGs (*p* < 0.05).

Taken together, among the 15 dysregulated genes, 9 up-regulated DEGs (*BIRC5*, *CDCA3*, *CENPF*, *KIF4A*, *NCAPG*, *RRM2*, *UBE2C*, *VEGFA*, and *NR2F6*) were significantly enriched in cell cycle regulation. Moreover, higher expression of *CDCA3* and *UBE2C* was found to be associated with poor overall survival of patients in all stages and a higher histologic grade. Thus, this bioinformatic analysis using various cohorts provides useful information of prognostic biomarkers, which could be effective targets for patients with serous ovarian cancer.

## 2. Materials and Methods

### 2.1. Microarray Data Source and Cluster Analysis

The 3 ovarian cancer-gene expression microarray datasets used in this study were obtained from the publicly available GEO database (National Institutes of Health, Bethesda, MD, USA; http://www.ncbi.nlm.nih.gov/geo/, date last accessed on 4 January 2021): GSE14407 [13], GSE36668 [14], and GSE38666 [15] (Table 1). The datasets were generated using the GPL570 platform (GeneChip Human Genome U133 Plus 2.0 Array; Affymetrix, Santa Clara, CA, USA). GSE14407 comprised 12 normal ovarian surface epithelial samples and 12 laser capture microdissected serous ovarian cancer epithelial samples, and GSE36668 comprised four surface epithelium scrapings from normal ovary samples, four serous ovarian borderline tumor samples, and four serous ovarian carcinoma samples, of which the normal and carcinoma of ovarian samples were herein analyzed. Furthermore, GSE38666 dataset comprised 45 samples consisting of eight normal stromal samples and eight matched normal ovarian surface epithelial samples from 12 individuals, along with seven cancer stromal samples and seven laser capture microdissected matched cancer epithelia from 18 ovarian cancer patients. Among these samples, data from 12 normal ovarian surface epithelial samples and 18 cancer epithelia from ovarian cancer patients were used for analysis. Cluster analysis was performed using Cluster 3.0 software [16] to classify the samples into statistically similar groups, and the resulting heatmaps were visualized using TreeView (version 1.6) [17]. The expression levels in heatmaps were log2 transformed, median-centered, and scaled for visualization.

### 2.2. Screening for DEGs

DEGs between normal ovarian surface epithelia and ovarian cancer epithelia were screened using the online analysis tool GEO2R (http://www.ncbi.nlm.nih.gov/geo/geo2r/, date last accessed on 25 January 2021) based on an adjusted *p*-value of less than 0.05. The top 250 significantly dysregulated genes were screened from each dataset regardless of positive or negative value of log2 fold change (FC) in the order of the lowest to the highest adjusted *p*-values. Common genes among the screened DEGs from the three GEO datasets were identified using Whitehead BaRC public tools (http://barc.wi.mit.edu/tools/, date last accessed on 28 January 2021). Expressions of the DEGs between normal and tumor tissue were obtained from TNMplot.com (http://tnmplot.com, date last accessed on 15 February 2021) [18], which contains microarray data from GEO database and RNA-sequencing data from TCGA, the Therapeutically Applicable Research to Generate Effective Treatments (TARGET), and the Genotype-Tissue Expression (GTEx) repositories. In the present study, RNA-seq data from TCGA and GTEx were used to compare the expression of the identified DEGs between ovarian serous cystadenocarcinoma patients and normal samples from non-cancerous patients.

### 2.3. GO Analysis of DEGs

The biological implications of the DEGs were analyzed with the online software The Database for Annotation, Visualization, and Integrated Discovery (DAVID) v6.8 (http://david.ncifcrf.gov/, date last accessed on 22 February 2021) [19]. Analyzed GO terms were subcategorized into biological process (BP), cellular component (CC), and molecular function (MF). *p* < 0.05 was considered statistically significant.

### 2.4. PPI Network Analysis

To analyze and visualize the functional PPI network of the identified DEGs, Cytoscape software (version 3.8.2) and Search Tool for the Retrieval of Interacting Genes (STRING, http://string-db.org, date last accessed on 8 March 2021) were used (maximum number of interactors = 0, and confidence score ≥ 0.9). The Cytoscape plugin Molecular Complex Detection (MCODE) was used to identify clustered modules within the PPI network (degree cutoff = 2, node score cutoff = 0.2, k-score = 2 and max. depth = 100).

### 2.5. DEG Survival Analysis

The Kaplan–Meier plotter (KM plotter, http://kmplot.com/, date last accessed on 22 March 2021) database contains gene expression and survival data from 1656 ovarian cancer patients from TCGA and GEO database. The dataset from GEO database includes the following: GSE3149, GSE9891, GSE14764, GSE15622, GSE18520, GSE19829, GSE23554, GSE26193, GSE26712, GSE27651, GSE30161, GSE51373, GSE63885, and GSE65986. In this study, overall survival data of 1207 ovarian cancer patients with serous histologic type were analyzed. The patients were grouped into higher and lower expression groups by dividing them at a cut-off value for the median expression of each DEG to give an “above-median to a higher expression” group and a “below-median to a lower expression” group. The association between the mRNA expression level of each DEG and the OS was analyzed using the Kaplan–Meier plotter. The hazard ratio (HR), 95% confidence intervals (CI), and the log-rank *p*-values were obtained from the website. A *p*-value < 0.05 was considered statistically significant.

### 2.6. Data Source for Analysis of Association with Clinicopathological Parameters

Prior to the analysis of association with clinicopathological features, the patients were grouped into higher and lower expression groups by dividing them at a cut-off value for the median expression of each DEG to give an “above-median to a higher expression” group and a “below-median to a lower expression” group. the RNA-seq gene expression dataset (dataset ID: TCGA.OV.sampleMap/HiseqV2) and the clinicopathological parameters (dataset ID: TCGA.OV.sampleMap/OV_clinicalMatrix) for the TCGA OV were downloaded from the UCSC Xena Browser (https://xena.ucsc.edu/, date last accessed on 29 March 2021). Patients without sufficient survival and gene expression data were excluded. The evaluated TCGA OV RNA-seq dataset comprised 308 samples, including 304 primary tumor tissue and 4 normal solid tissue samples. Additionally, the microarray dataset (dataset ID: TCGA.OV.sampleMap/HT_HG-U133A) was downloaded for the analysis of gene expression level according to histologic grade. TCGA OV microarray dataset comprised 593 samples. In this data, gene expression was measured using Affymetrix HT Human Genome U133a microarray platform. Gene expression level in this data is in log2(x). Patients without sufficient information of histologic grade were excluded. This study met the publication guidelines provided by TCGA (https://www.cancer.gov/about-nci/organization/ccg/research/structural-genomics/tcga/using-tcga/citing-tcga, date last accessed on 31 March 2021).

### 2.7. Statistical Analysis

SPSS version 26.0 for Windows (SPSS Inc, Chicago, IL, USA) was used to analyze the collected data. The difference between gene expression and clinical information was analyzed by performing the Pearson’s chi-squared test for categorical variables. The Student’s *t*-test was used to analyze the difference between patients with histologic grades 1 or 2 and those with histologic grades 3 or 4. A *p*-value less than 0.05 was considered statistically significant.

## 3. Results

### 3.1. DEG Identification

Screening by GEO2R analysis revealed 54,660, 52,001, and 54,666 DEGs from the GSE14407, GSE36668, and GSE38666 datasets, respectively. Then, Venn diagram software was used for identifying the common DEGs among the three datasets. Overall, 15 common DEGs were identified (Figure 1), including nine up-regulated and six down-regulated genes in serous ovarian cancer tissues compared with normal ovarian tissues (Figure 2 and Table 2 and Table 3). LOC101930363/LOC101928349/LOC100507387/FAM153C/FAM153A/FAM153B, among the down-regulated DEGs, was not evaluated further due to insufficient information on the gene.

### 3.2. Expression of the Identified DEGs between Normal and Serous Ovarian Cancer Tissues

The expression of the identified DEGs between normal and tumor tissues was analyzed using RNA–seq data from TNMplot.com. The up–regulated DEGs showed significantly higher expression in tumor tissues compared with normal tissues (Figure 3, Table 4), whereas the down-regulated DEGs showed significantly higher expression in normal tissues compared with tumor tissues (Figure 4, Table 5).

### 3.3. PPI Network Construction

The PPI network was analyzed and visualized using Cytoscape software. Using the STRING database and MCODE tool, among the 14 DEGs shared by the 3 GEO datasets, we identified one cluster with 7 nodes and 21 edges (Figure 5). The clustered genes were *BIRC5*, *CDCA3*, *NCAPG*, *UBE2C*, *KIF4A*, *CENPF*, and *RRM2*, which were all up-regulated DEGs. *CDCA3* was identified as the seed node from which the cluster was derived.

### 3.4. Clustered Genes Are Mainly Involved in Cell Cycle Regulation

To analyze the functional profile of the clustered DEGs, GO term analysis was performed using DAVID. The up–regulated DEGs, especially the clustered DEGs, were mainly involved in cell division, positive regulation of exit from mitosis, and mitotic nuclear division, whereas the down-regulated DEGs were not functionally clustered, with exception of *RORA*, which was involved in the intracellular receptor signaling and steroid hormone–mediated signaling pathways (Table 6).

### 3.5. Higher Expression of CDCA3 and UBE2C Associated with Poor OS in All Stages

Prior to the Kaplan–Meier survival analysis using the Kaplan–Meier plotter, all patients with serous ovarian cancer were categorized into disease stages as early (including stage I and II) and advanced (including stage III and IV) stages. This showed that higher expressions of *CDCA3*, *CENPF*, *NCAPG*, *RRM2*, *UBE2C*, and *NBEA* were associated with worse OS regardless of serous ovarian cancer stages (Figure 6). Moreover, in patients with early stages, higher expressions of *CDCA3*, *CENPF*, *NCAPG*, *N2RF6*, *RRM2*, *UBE2C*, and *NBEA* were associated with worse OS (Figure 7). However, in the advanced stages, only higher expressions of *CDCA3*, *NR2F6*, and *UBE2C* were associated with worse OS, whereas higher expression of *BIRC5* was associated with better OS (Figure 8).

### 3.6. Higher CDCA3 and UBE2C Expression Associated with Higher Histologic Grades

Based on data from the TCGA OV cohort, we analyzed the association between expressions of DEGs and clinicopathological features of serous ovarian cancer. Higher expressions of *CDCA3* and *UBE2C* were significantly associated with higher histologic grades, but not with other clinical features, including age, clinical stage, lymphatic invasion, venous invasion, primary therapy outcome, personal tumor status, and vital status (Figure 9 and Appendix A).

## 4. Discussion

Ovarian cancer is one of the leading causes of cancer death in women’s reproductive organs [1]. According to the Surveillance, Epidemiology, and End Results Program: Ovarian Cancer (http://seer.cancer.gov/, date last accessed on 5 July 2021), in 2021, approximately 21,410 patients are expected to be diagnosed, and 13,770 are estimated to die of ovarian cancer in the United States [20]. Approximately 70% of newly diagnosed ovarian cancer patients are diagnosed in advanced stages [2]. Most patients, even those with advanced stage cancer, show efficient treatment outcomes with primary treatment, which is generally primary debulking surgery followed by taxane/platinum-based chemotherapy [21]. Nevertheless, up to 80% of these patients experience cancer recurrence, which is associated with extremely poor survival outcomes [22]. There are currently established factors for predicting recurrence and prognosis including age, clinical stages, histologic grades, and size of residual diseases after surgery [23,24]. Despite these prognostic factors, heterogenic outcome of ovarian cancer to standard treatment makes these factors less reliable, which suggests finding more reliable factors for predicting prognosis of ovarian cancer might provide clinical value.

Approximately 90% of ovarian cancers are epithelial ovarian cancer, among which the vast majority are serous ovarian cancers [25]. Although the ovarian cancer developmental mechanism is not fully understood, numerous bioinformatics tools such as microarray and high-throughput sequencing have been developed over the past few decades, enabling the discovery of biomarkers for diagnosis and prognosis. In the present study, we focused on identifying the common DEGs based on microarray data from three GEO datasets. By finding common DEGs between normal and tumor tissues with the most significant FC from within each GEO dataset, we identified nine up-regulated DEGs and five down-regulated DEGs.

The PPI network of 14 DEGs revealed one cluster that comprised the seven most functionally connected genes (*BIRC5*, *CDCA3*, *CENPF*, *KIF4A*, *NCAPG*, *RRM2*, and *UBE2C)*, which were all up–regulated DEGs. Among the clustered genes, *CDCA3* was identified to act as a key component in this cluster. GO term analysis of the 14 DEGs further showed that the up-regulated DEGs, especially the clustered genes identified in the PPI network, were mainly involved in the biological processes of cell division, positive regulation of exit from mitosis, and mitotic nuclear division. Uncontrolled cell cycle is a common feature of cancer, and it has been a main therapeutic target to impair cancer cell proliferation. However, the prognostic value of aberrant expression of genes involved in the regulation of cell cycle in ovarian cancer is still unclear.

To verify the prognostic value of the identified DEGs, Kaplan–Meier survival analysis was performed based on the 14 DEGs. Overall, five genes among the down–regulated DEGs were not associated with favorable outcome in OS, but five out of the nine up–regulated DEGs were associated with worse survival regardless of the stage of serous ovarian cancer, and six genes were associated with worse survival in early stage (stage I and II) of ovarian cancer. Remarkably, higher expression of *BIRC5* was associated with better OS in the advanced stages of serous ovarian cancer. Additionally, higher expression of NBEA was associated with worse OS in the early stages of serous ovarian cancer. However, *BIRC5* and *NBEA* did not show a significant difference in clinical-pathologic features (data not shown). In absence of additional supporting evidence, it is not possible to conclude that dysregulation of *BIRC5* and *NBEA* affects the prognosis of serous ovarian cancer. Interestingly, while the associations between OS and some DEGs were not significant or were even ambivalent, *CDCA3* and *UBE2C*, which were among the up–regulated DEGs, were found to be consistently associated with worse survival in all stages, early stages and advanced stages. Moreover, a higher expression of the *CDCA3* and *UBE2C* genes was significantly associated with higher tumor histologic grades. Of note, *CDCA3* was identified as a key component among the clustered genes within the DEGs. Ubiquitin-conjugating enzyme, E2C, is a member of the UBE2 family that mediates cell cycle progression and mitotic cyclin destruction. Recent studies showed that overexpression of *UBE2C* is associated with various cancers such as breast, colon, lung, and liver cancers. It was also reported that the aberrant expression of *UBE2C* is associated with some gynecologic cancers, such as endometrial and ovarian cancers. Some studies suggested *UBE2C* as a prognostic biomarker and potential therapeutic target in certain cancers. For example, Zhang et al. suggested *UBE2C* as a prognostic marker and a promising therapeutic target in gastric cancer [26], and Li et al. reported *UBE2C* as a prognostic biomarker and a potential therapeutic target associated with chemo–resistance in ovarian cancer [27]. In the contrast, little is known about the association of *CDCA3* and ovarian cancer. *CDCA3* cell–division–cycle-associated protein–3 is a member of the CDCA family and modulates mitosis entry and cell cycle, which is regulated by protein degradation during the G1 phase [28]. Adams et al. suggested *CDCA3* as a prognostic biomarker and potential therapeutic target in non-small–cell lung cancer [29], and Zhang et al. suggested *CDCA3* as a potential prognostic marker in gastric cancer [30]. In the present study, we identified 14 DEGs from GEO datasets. Among them, we found nine up–regulated DEGs and seven clustered genes that were mainly associated with cell division and mitosis. By analysis of the clinical data, we found that *CDCA3* and *UBE2C* were associated with poor prognosis in serous ovarian cancer patients. Further studies are necessary to verify the prognostic value of *CDCA3* and *UBE2C* and to further understand the role of these genes in ovarian cancer to establish them as potential therapeutic targets.

## 5. Conclusions

Based on the above-mentioned findings, *CDCA3* and *UBE2C* were identified as being associated with higher tumor histologic grades and poor survival outcomes in patients with serous ovarian cancer. We suggested that *CDCA3* and *UBE2C* may represent valuable biomarkers to predict the outcome of serous ovarian cancer. Nevertheless, further investigations are warranted to obtain a better understanding of their function in serous ovarian cancer.

## Figures and Tables

**Figure 1 medicina-57-00933-f001:**
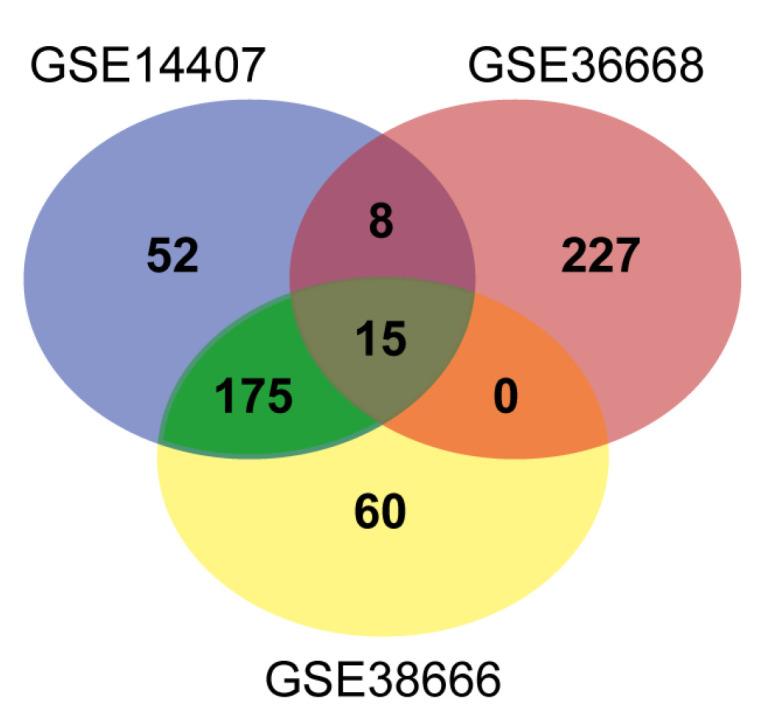
Venn diagram of DEGs among the GEO datasets.

**Figure 2 medicina-57-00933-f002:**
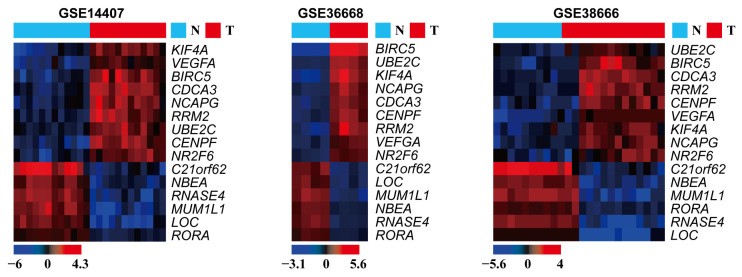
Heatmap showing the 15 identified DEGs in the 3 independent GEO datasets. The data are presented in matrix format, with rows representing individual genes and columns representing each tissue. Each cell in the matrix represents the relative expression level of a genetic characteristic of an individual tissue. The red and green colors in the cells reflect relatively high and low expression levels, respectively, as indicated by the scale bar. N: normal; T: tumor; LOC: LOC101930363/LOC101928349/LOC100507387/FAM153C/FAM153A/FAM153B.

**Figure 3 medicina-57-00933-f003:**
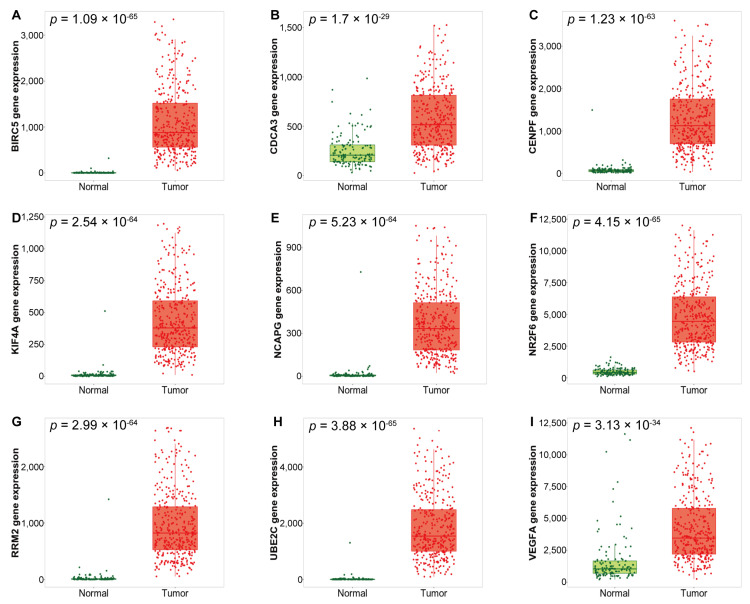
Box plots of up-regulated DEGs expression between normal and serous ovarian cancer tissues in the RNA-seq dataset of TCGA OV. (**A**) *BIRC5*, (**B**) *CDCA3*, (**C**) *CENPF*, (**D**) *KIF4A*, (**E**) *NCAPG*, (**F**) *NR2F6*, (**G**) *RRM2*, (**H**) *UBE2C*, and (**I**) *VEGFA* mRNA expression.

**Figure 4 medicina-57-00933-f004:**
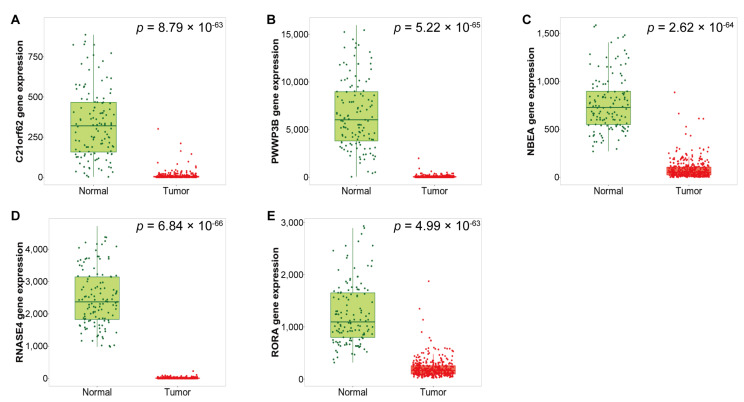
Box plots of down-regulated DEGs expression between normal and serous ovarian cancer tissues in the RNA-seq dataset of TCGA OV. (**A**) *C21orf62*, (**B**) *PWWP3B*, (**C**) *NBEA*, (**D**) *RNASE4*, and (**E**) *RORA* mRNA expression.

**Figure 5 medicina-57-00933-f005:**
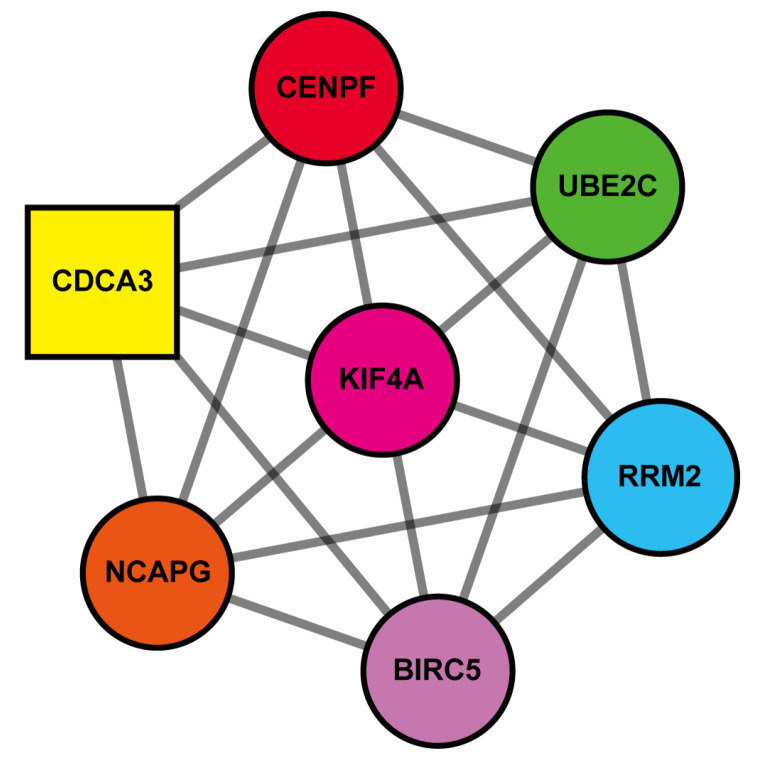
Clustered DEGs identified by using MCODE in PPI network. Seven nodes and twenty–one edges were identified. Square shape node is a seed node, and other nodes have same score.

**Figure 6 medicina-57-00933-f006:**
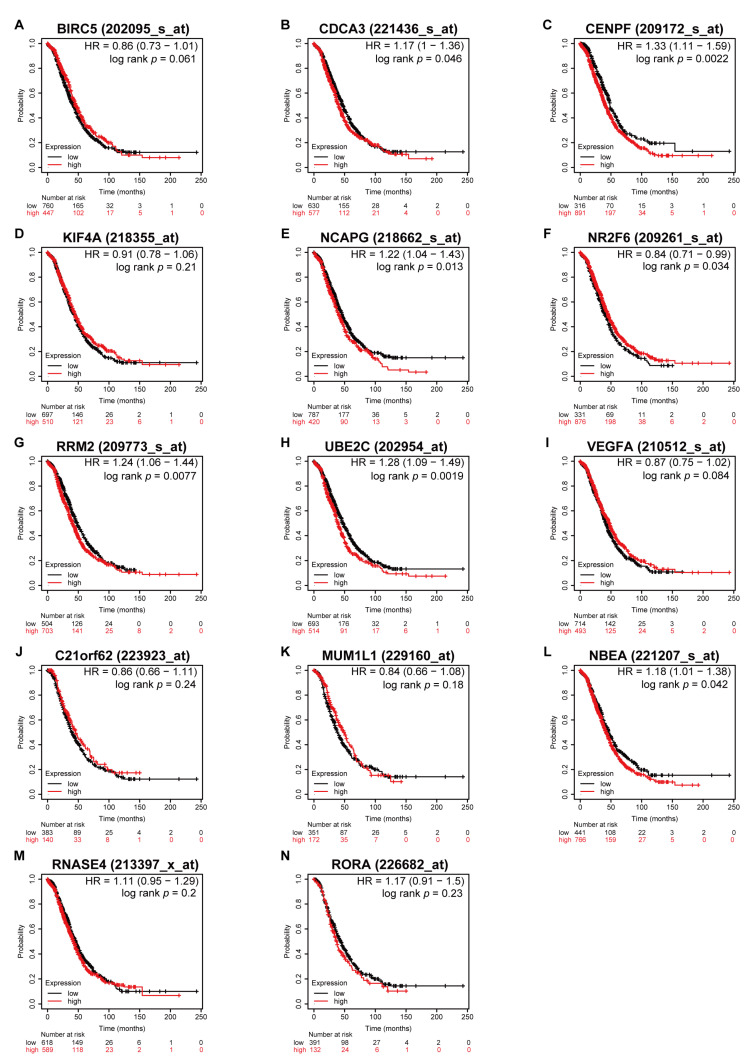
Survival analyses of the dysregulated genes in all stages of serous ovarian cancer. Kaplan–Meier plots of overall survival in subjects with higher versus lower expression. (**A**) *BIRC5*, (**B**) *CDCA3*, (**C**) *CENPF*, (**D**) *KIF4A*, (**E**) *NCAPG*, (**F**) *NR2F6*, (**G**) *RRM2*, (**H**) *UBE2C*, (**I**) *VEGFA*, (**J**) *C21orf62*, (**K**) *MUM1L1*, (**L**) *NBEA*, (**M**) *RNASE4*, and (**N**) *RORA* mRNA expression.

**Figure 7 medicina-57-00933-f007:**
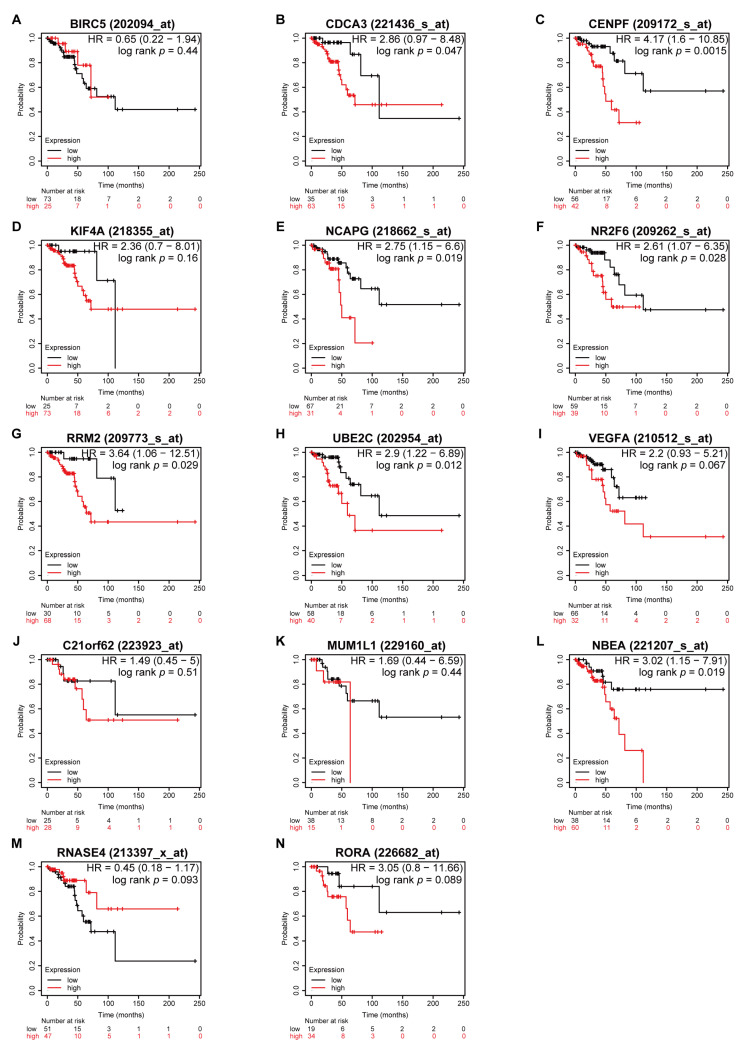
Survival analyses of the dysregulated genes in the early stages of serous ovarian cancer. Kaplan–Meier plots of overall survival in subjects with higher versus lower expression. (**A**) *BIRC5*, (**B**) *CDCA3*, (**C**) *CENPF*, (**D**) *KIF4A*, (**E**) *NCAPG*, (**F**) *NR2F6*, (**G**) *RRM2*, (**H**) *UBE2C*, (**I**) *VEGFA*, (**J**) *C21orf62*, (**K**) *MUM1L1*, (**L**) *NBEA*, (**M**) *RNASE4*, and (**N**) *RORA* mRNA expression.

**Figure 8 medicina-57-00933-f008:**
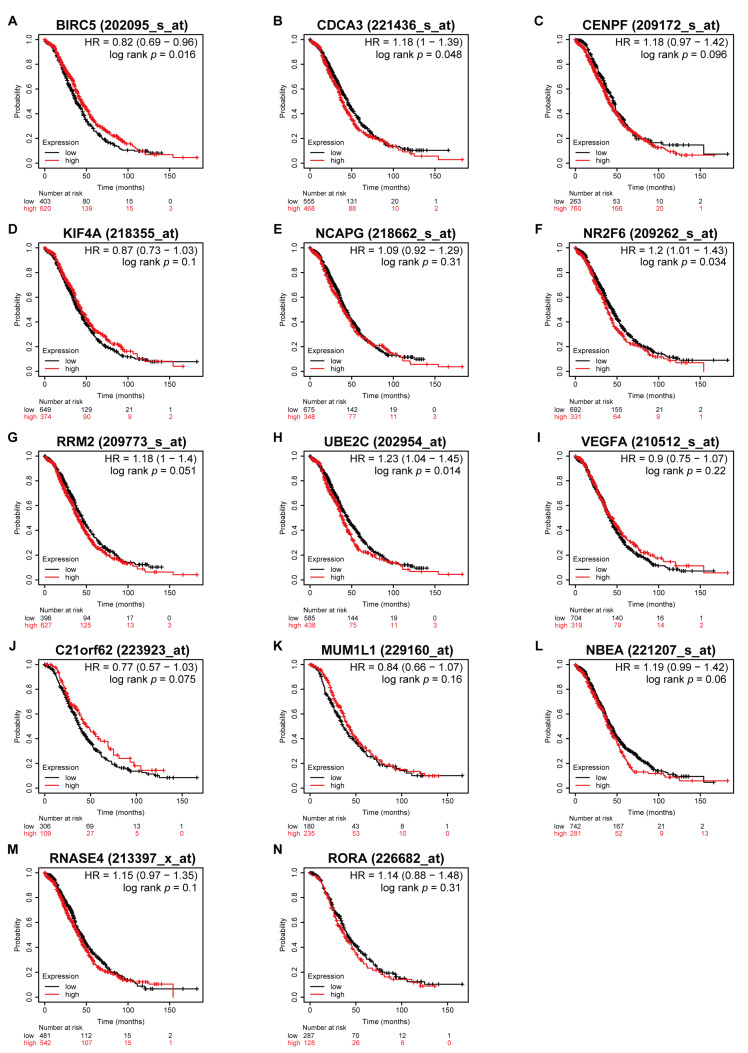
Survival analyses of the dysregulated genes in the advanced stages of serous ovarian cancer. Kaplan–Meier plots of overall survival in subjects with higher versus lower expression. (**A**) *BIRC5*, (**B**) *CDCA3*, (**C**) *CENPF*, (**D**) *KIF4A*, (**E**) *NCAPG*, (**F**) *NR2F6*, (**G**) *RRM2*, (**H**) *UBE2C*, (**I**) *VEGFA*, (**J**) *C21orf62*, (**K**) *MUM1L1*, (**L**) *NBEA*, (**M**) *RNASE4*, and (**N**) *RORA* mRNA expression.

**Figure 9 medicina-57-00933-f009:**
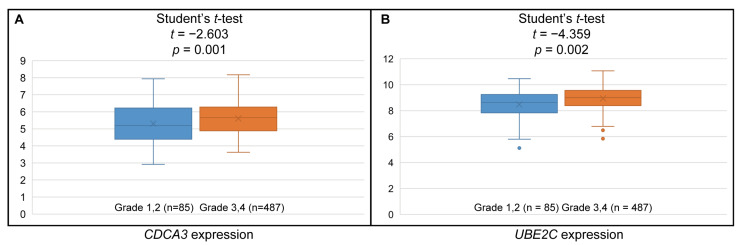
Higher expression levels of *CDCA3* and *UBE2C* are associated with higher histologic grade of tumor. (**A**) Box plot of the relative *CDCA3* expression levels and (**B**) *UBE2C* expression levels in the grade 1 and 2 and grade 3 and 4 serous ovarian cancer tissues.

**Table 1 medicina-57-00933-t001:** Basic information of microarray data from NCBI GEO database.

Platform	GEO Dataset	Samples	Reference
GPL570	GSE14407	12 Normal, 12 Cancer	Bowen et al. [13]
GPL570	GSE36668	4 Normal, 4 Cancer	Elgaaen et al. [14]
GPL570	GSE38666	12 Normal, 18 Cancer	Lili et al. [15]

NCBI: The National Center for Biotechnology Information; GEO: Gene Expression Omnibus.

**Table 2 medicina-57-00933-t002:** Up-regulated DEGs in serous ovarian cancer of the GEO datasets.

Gene Names	Log2FC			Adjusted *p*-Value		
	GSE14407	GSE36668	GSE38666	GSE14407	GSE36668	GSE38666
*BIRC5*	3.784515	13.83283	3.792818	4.58 × 10^−6^	2.45 × 10^−4^	3.34 × 10^−8^
*CDCA3*	3.40002	3.542135	3.543772	9.12 × 10^−7^	2.72 × 10^−3^	2.20 × 10^−9^
*CENPF*	3.904451	3.450972	3.934447	1.69 × 10^−7^	3.17 × 10^−3^	2.20 × 10^−9^
*KIF4A*	3.533572	5.173891	3.746773	4.76 × 10^−6^	2.84 × 10^−3^	1.70 × 10^−8^
*NCAPG*	3.233566	3.533272	3.092676	5.15 × 10^−6^	3.17 × 10^−3^	1.26 × 10^−7^
*RRM2*	3.212708	4.504844	3.215252	3.24 × 10^−6^	3.17 × 10^−3^	2.23 × 10^−8^
*UBE2C*	2.539233	3.816594	2.699924	7.37 × 10^−6^	2.56 × 10^−3^	1.26 × 10^−7^
*VEGFA*	2.86746	2.906894	2.647287	2.89 × 10^−6^	2.84 × 10^−3^	4.33 × 10^−7^
*NR2F6*	2.950696	2.75339	3.029833	1.85 × 10^−5^	2.72 × 10^−3^	1.63 × 10^−7^

DEGs: differentially expressed genes; FC: fold change.

**Table 3 medicina-57-00933-t003:** Down-regulated DEGs in serous ovarian cancer of the GEO datasets.

Gene Names	Log2FC			Adjusted *p*-Value		
	GSE14407	GSE36668	GSE38666	GSE14407	GSE36668	GSE38666
*C21orf62*	−3.784515	−13.83283	−3.792818	4.58 × 10^−6^	2.45 × 10^−4^	3.34 × 10^−8^
*MUM1L1*	−3.40002	−3.542135	−3.543772	9.12 × 10^−7^	2.72 × 10^−3^	2.20 × 10^−9^
*NBEA*	−3.904451	−3.450972	−3.934447	1.69 × 10^−7^	3.17 × 10^−3^	2.20 × 10^−9^
*RNASE4*	−3.533572	−5.173891	−3.746773	4.76 × 10^−6^	2.84 × 10^−3^	1.70 × 10^−8^
*RORA*	−3.233566	−3.533272	−3.092676	5.15 × 10^−6^	3.17 × 10^−3^	1.26 × 10^−7^
*LOC101930363* */LOC101928349* */LOC100507387* */FAM153C* */FAM153A* */FAM153B*	−4.26894	−3.17681	−4.17039	4.56 × 10^−7^	0.003168	8.62 × 10^−9^

DEGs: differentially expressed genes; FC: fold change.

**Table 4 medicina-57-00933-t004:** Up-regulated DEGs in serous ovarian cancer of the RNA-seq dataset of TCGA OV.

Gene Names	Fold Change	*p*-Value
*BIRC5*	157.01	1.09 × 10^−65^
*CDCA3*	2.58	1.7 × 10^−29^
*CENPF*	18.82	1.23 × 10^−63^
*KIF4A*	34.68	2.54 × 10^−64^
*NCAPG*	29.63	5.23 × 10^−64^
*NR2F6*	11.6	4.15 × 10^−65^
*RRM2*	41.23	2.99 × 10^−64^
*UBE2C*	101.05	3.88 × 10^−65^
*VEGFA*	2.61	3.13 × 10^−34^

DEGs: differentially expressed genes.

**Table 5 medicina-57-00933-t005:** Down-regulated DEGs in serous ovarian cancer of the RNA-seq dataset of TCGA OV.

Gene Names	Fold Change	*p*-Value
*C21orf62*	0.02	8.79 × 10^−63^
*MUM1L1(PWWP3B)*	0.01	5.22 × 10^−65^
*NBEA*	0.11	2.62 × 10^−64^
*RNASE4*	0.01	6.84 × 10^−66^
*RORA*	0.15	4.99 × 10^−63^

DEGs: differentially expressed genes.

**Table 6 medicina-57-00933-t006:** GO terms of DEGs in serous ovarian cancer.

Category	Term	Gene Names	*p*-Value
GO_BP	GO:0051301 Cell division	*BIRC5*, *CDCA3*, *CENPF*, *NCAPG*, *UBE2C*	5.5 × 10^−5^
	GO:0031536 Positive regulation of exit from mitosis	*BIRC5*, *UBE2C*	3.9 × 10^−3^
	GO:0000278 Mitotic nuclear division	*BIRC5*, *CDCA3*, *CENPF*	1.1 × 10^−2^
	GO:0016567 Protein ubiquitination	*BIRC5*, *CDCA3*, *UBE2C*	2.2 × 10^−2^
	GO:0030522 Intracellular receptor signaling pathway	*RORA*, *NR2F6*	2.5 × 10^−2^
	GO:0043401 Steroid hormone mediated signaling pathway	*RORA*, *NR2F6*	3.7 × 10^−2^
	GO:0043154 Negative regulation of cysteine–type endopeptidase activity involved in apoptotic process	*BIRC5*, *VEGFA*	4.4 × 10^−2^
GO_CC	Cytosol	*BIRC5*, *CDCA3*, *CENPF*, *KIF4A*, *NBEA*, *NCAPG*, *RRM2*, *UBE2C*	2.2 × 10^−3^
	Midbody	*BIRC5*, *CENPF*, *KIF4A*	3.1 × 10^−3^
	Nucleoplasm	*RORA*, *BIRC5*, *CENPF*, *KIF4A*, *NR2F6*, *RRM2*, *UBE2C*	5.1 × 10^−3^
	Spindle microtubule	*BIRC5*, *KIF4A*	2.9 × 10^−2^
	Chromosome, centromeric region	*BIRC5*, *CENPF*	3.7 × 10^−2^
GO_MF	RNA polymerase II transcription factor activity, ligand activated sequence–specific DNA	*RORA*, *NR2F6*	2.3 × 10^−2^
	Steroid hormone receptor activity	*RORA*, *NR2F6*	3.6 × 10^−2^
	Protein binding	*RORA*, *BIRC5*, *CDCA3*, *CENPF*, *KIF4A*, *NCAPG*, *NR2F6*, *RRM2*, *UBE2C*, *VEGFA*	4.4 × 10^−2^

GO, gene ontology; BP, biological process; CC, cellular component; MF, molecular function.

## Data Availability

All available data are presented within the article or are available from the corresponding authors upon adequate request.

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
