# Peer review of "Bioinformatic Analysis for Influential Core Gene Identification and Prognostic Significance in Advanced Serous Ovarian Carcinoma"

_medicina, 2021, doi:10.3390/medicina57090933_

Round 1

Reviewer 1 Report

The paper by Song et al compares 3 microarray datasets comparing normal ovary surface epithelia to ovarian cancer surface epithelia from the Gene Expression Omnibus to identify possible biomarkers for ovarian cancer.  From this analysis, they identified only nine upregulated genes and six downregulated genes shared across the three datasets. These genes were also upregulated compared to normal samples in a separate RNA-seq dataset. They find that Seven of the nine upregulated genes are clustered based on protein-protein interactions and are involved in cell cycle regulation. The authors then analyzed the correlation between high or low expression of these genes and survival based on RNA-seq analysis of 304 ovarian cancer samples in the TCGA that are linked to survival data. They show that higher expression of six of the nine upregulated genes is associated with worse overall survival.  Finally, they focus on two genes, CDCA3 and UBE2C, associated with poor prognosis and higher tumor histologic grade.

This study is interesting because it compares data from multiple studies and finds very few genes that show similar responses.  This may be due to the heterogeneity of ovarian cancer or it may reflect differences in study design. It is not surprising that the upregulated genes compared to normal ovary epithelia, were clustered around cell cycle regulation. However, it is interesting that these genes are also associated with poor overall survival when tested on an RNA-seq data set with far more samples.

Major comments:

1. In the paper, the authors should emphasize that they used microarray for the original analysis, but then examined expression of the genes of interest in RNA-seq data sets. This makes the analysis more robust. In the Results, it should be clarified in the Figure legends and Tables (Tables 2, 3, 4 and 5, Figures 3 and 4) whether the analysis is from the original microarray datasets or from the RNA-Seq data set. This will better highlight these two analyses.

2. The authors need to better present previous work on ovarian cancer biomarkers in the Introduction and show how their work is forwarding the study of ovarian cancer biomarkers. Were these 15 genes identified in the original studies as significantly upregulated in the ovarian cancer samples?  What is the power of this meta-analysis? 

3. It is interesting that all three studies have only 15 genes in common. There should be more discussion of this and whether this was expected. One might expect many more changes in common when comparing normal tissue and ovarian cancer samples. Does this speak to the heterogeneity of ovarian cancer?

4. The Discussion should include not only details about CDCA3 and UBE2C, but also about the other upregulated genes that also showed a correlation with prognosis. Discuss why BIRC5 appears to be correlated with better prognosis.  NBEA seems to be spectacularly different in early stage cancer.  Is there any evidence for how these other genes might influence prognosis?

5. In general, the grammar needs editing especially in the Introduction and Discussion. Example: “Most ovarian cancers are symptomless which makes up to 70% of ovarian cancer patients are diagnosed in advanced stage…” (Page 1 line 44) This should be “Most ovarian cancers are symptomless which MEANS THAT up to 70% of ovarian cancer patients are diagnosed AT AN advanced stage…”

Minor comments:

1. The text states that the top 250 genes with significant change in expression were compared (Page 3 line 26), but the Venn diagram (Fig 1) has 280 genes per study. Please fix this inconsistency.

2. Clarify what “serous ovarian cancer epithelial samples” (Page 2 line 78) and “matched cancer epithelia” (Page 2, line84) means. Does this mean epithelial ovarian cancer samples? Are they from bulk tumor?

3. Figures 1 and 2 use red and green that is problematic for color blind readers. This should be switched to magenta and green.

4. Page 9 paragraph 3.5, the authors talk about subdividing the samples into stage 1,2 and stage 3,4, then talk about higher expression of certain genes being associated with worse overall survival in all stages. This was confusing. The authors should first talk about associations in all stages, then describe how they were categorized as early or late stage and then show results from the two categories.

5. Figure 6 has writing in a very tiny font that is illegible. The font needs to be increased.

6. The first line of the Discussion says that ovarian cancer is one of the leading causes of death among females. This should be toned down. In female cancer deaths, ovarian cancer ranks fifth.

7. Page 12, line 234, the authors state, “Higher CDCA3 and UBE2C gene expression was significantly associated with higher histologic grades, while neither gene was significantly associated with other clinical features.” Please list the other clinical features that were analyzed.    

Author Response

August 23rd, 2021

Dear Reviewer 1,

Bioinformatic analysis for influential core gene identification and prognostic significance in advanced serous ovarian carcinoma

     We sincerely appreciate the time and effort of you and the referees spent in considering and evaluating our manuscript (medicina-1325979) entitled with “Bioinformatic analysis for influential core gene identification and prognostic significance in advanced serous ovarian carcinoma” for publication in the Medicina. Having received the kind comments by you, we revised our manuscript with attention to each of the comments by you. We appreciate the reviewer 1 very much, who raised the very important critiques to strengthen the claim of our manuscript. We have given very careful consideration to the suggestions and have revised our manuscript. We incorporated new information in the revised version of our manuscript.

     We have fully described the manuscript addressing to the particular issue raised. We have responded all the comments by the reviewer 1 point-by-point as follows. We marked light blue on the corrected or added parts on the manuscript in comparison with un-revised manuscript.

     Thank you very much in advance for your time and effort involved in arranging review process.

Yours Sincerely,

Shin Kim, M. D., Ph. D.

The corresponding author

Reviewer’s comments

# Reviewer 1

The paper by Song et al compares 3 microarray datasets comparing normal ovary surface epithelia to ovarian cancer surface epithelia from the Gene Expression Omnibus to identify possible biomarkers for ovarian cancer.  From this analysis, they identified only nine upregulated genes and six downregulated genes shared across the three datasets. These genes were also upregulated compared to normal samples in a separate RNA-seq dataset. They find that Seven of the nine upregulated genes are clustered based on protein-protein interactions and are involved in cell cycle regulation. The authors then analyzed the correlation between high or low expression of these genes and survival based on RNA-seq analysis of 304 ovarian cancer samples in the TCGA that are linked to survival data. They show that higher expression of six of the nine upregulated genes is associated with worse overall survival.  Finally, they focus on two genes, CDCA3 and UBE2C, associated with poor prognosis and higher tumor histologic grade.

This study is interesting because it compares data from multiple studies and finds very few genes that show similar responses.  This may be due to the heterogeneity of ovarian cancer or it may reflect differences in study design. It is not surprising that the upregulated genes compared to normal ovary epithelia, were clustered around cell cycle regulation. However, it is interesting that these genes are also associated with poor overall survival when tested on an RNA-seq data set with far more samples.

Major comments:

  1. In the paper, the authors should emphasize that they used microarray for the original analysis, but then examined expression of the genes of interest in RNA-seq data sets. This makes the analysis more robust. In the Results, it should be clarified in the Figure legends and Tables (Tables 2, 3, 4 and 5, Figures 3 and 4) whether the analysis is from the original microarray datasets or from the RNA-Seq data set. This will better highlight these two analyses.

Answer: Thank you for your kind advice. As you mentioned, we clarified the information of datasets which are used in the Figures and Tables. As shown in Table 2, 3, 4, and 5, we described in detail for distinguishing whether RNA-seq data sets or GEO dataset were used (page 5, line 202; page 6 line 205; page 7, line 218; page 8, line 225). Moreover, in the legends of both Figure 3 and 4, we corrected figure legends to clarify the information of datasets (page 7, line 216; page 8 line 223).

  1. The authors need to better present previous work on ovarian cancer biomarkers in the Introduction and show how their work is forwarding the study of ovarian cancer biomarkers. Were these 15 genes identified in the original studies as significantly upregulated in the ovarian cancer samples?  What is the power of this meta-analysis? 

Answer: Thank you for your kind advice. As you mentioned, we presented previous work on ovarian cancer biomarkers in the Introduction (references no. 6-8) and described how their work is forwarding the study on the therapeutic targets and prognostic markers of ovarian cancer in the Introduction (page 2, line 59). The 15 genes were dysregulated in all three GEO datasets (Figure 1, Table 2 and 3). As you know, meta-analysis is a statistical analysis that combines the results of multiple scientific studies. Actually, our study is not a meta-analysis, but a small-scale data reanalysis study using publically available gene expression databases such as Gene Expression Omnibus and The Cancer Genome Atlas cohorts. Moreover, we believe that meta-analysis of this reanalysis studies can be helpful in discovering diagnostic, therapeutic, and prognostic biomarkers for ovarian cancer.

  1. It is interesting that all three studies have only 15 genes in common. There should be more discussion of this and whether this was expected. One might expect many more changes in common when comparing normal tissue and ovarian cancer samples. Does this speak to the heterogeneity of ovarian cancer?

Answer: Thank you for your kind advice. Actually, after analysis of DEGs, each dataset contained DEGs over 50,000 Affymetrix gene ID samples. As you can see in DEG screening paragraph of our manuscript, among these differential expression data from each dataset, top 250 genes from the lowest p value were selected. The reason there was only 15 genes in common was we only compared top 250 DEGs from each dataset. Moreover, we agreed your opinion ‘one might expect many more changes in common when comparing normal tissue and ovarian cancer samples’. Actually, the normal tissue used in GEO datasets and the TCGA OV cohort was pathologically non-neoplastic tissues around the surgically removed ovarian cancer tumor. Although, as the reviewer commented, using normal tissues unrelated to cancer is a very good method of research, the study using paired samples of cancer and adjacent non-neoplastic tissues is to exclude bias caused by genetic background between samples. Additionally, single cell sequencing technology has been recently introduced because tumor heterogeneity can affect the result of gene expression of tumor. But only a few researchers have an opportunity of single cell sequencing because the cost of analysis is too high. Nevertheless, as a follow-up study, we’re planning a study using single cell sequencing.

  1. The Discussion should include not only details about CDCA3 and UBE2C, but also about the other upregulated genes that also showed a correlation with prognosis. Discuss why BIRC5 appears to be correlated with better prognosis.  NBEA seems to be spectacularly different in early stage cancer.  Is there any evidence for how these other genes might influence prognosis?

Answer: Thank you for your kind advice. In this study, we concluded CDCA3 and UBE2C are the only potential biomarkers for predicting ovarian cancer prognosis. This is because CDCA3 and UBE2C showed consistent association with higher expression in tumor tissue, higher histologic grade and poor survival outcome among stages. The relationship with higher histologic grade supports the association with poor survival outcome. On the other hand, BIRC5 was highly expressed in tumor but showed association with better survival in advanced stages of serous ovarian cancer patients. To validate prognostic value of BIRC5, we investigated relationship of clinic-pathologic features and relationship of survival. BIRC5 did not show significant difference in clinic-pathologic features and it was not associated with survival outcome except with advanced stages of ovarian cancer patients. Without supportive evidences, single aspect of showing better survival outcome in advanced stages within the population in this study cannot be generalized. Likewise, it is difficult to explain the unexpected result of BIRC5 and NBEA but we considered it as unremarkable result. As you mentioned, we described the information of BIRC5 and NBEA in the revised manuscript (page 15, line 338).

  1. In general, the grammar needs editing especially in the Introduction and Discussion. Example: “Most ovarian cancers are symptomless which makes up to 70% of ovarian cancer patients are diagnosed in advanced stage…” (Page 1 line 44) This should be “Most ovarian cancers are symptomless which MEANS THAT up to 70% of ovarian cancer patients are diagnosed AT AN advanced stage…”

Answer: Thank you for your kind advice. As you mentioned, we carefully read the sentence. Then we corrected the sentence to clarify the meaning of the description. You can see the corrected description in the revised manuscript (page 2, line 50; page 14, line 304).

Minor comments:

  1. The text states that the top 250 genes with significant change in expression were compared (Page 3 line 26), but the Venn diagram (Fig 1) has 280 genes per study. Please fix this inconsistency.

Answer: Thank you for your kind advice. As you mentioned, we corrected the Venn diagram (Figure 1). You can see the corrected Figure 1 in the revised manuscript.

  1. Clarify what “serous ovarian cancer epithelial samples” (Page 2 line 78) and “matched cancer epithelia” (Page 2, line84) means. Does this mean epithelial ovarian cancer samples? Are they from bulk tumor?

Answer: Thank you for your kind advice. As you mentioned, we carefully read the information of the GEO datasets. Serous ovarian cancer epithelial samples of GSE14407 are isolated by laser capture microdissection from human serous papillary ovarian adenocarcinomas. You can find the research design and method in the following study, ‘Bowen et al., Gene expression profiling supports the hypothesis that human ovarian surface epithelia are multipotent and capable of serving as ovarian cancer initiating cells. BMC Med Genomics. 2009.’ Moreover, Matched cancer epithelia of GSE38666 are obtained by lase capture microdissection from the tumors of 18 serous ovarian cancer patients. You can find the research design and method in the following study, ‘Lili et al., Molecular profiling predicts the existence of two functionally distinct classes of ovarian cancer stroma. Biomed Res Int. 2013.’ Based on these description, we corrected the description in the revised manuscript. You can see the corrected description (page 3, line 92 and 98).

  1. Figures 1 and 2 use red and green that is problematic for color blind readers. This should be switched to magenta and green.

Answer: Thank you for your kind advice. We feel sorry that we did not give concern to color blind readers. As you mentioned, we changed colors in figure 1 and 2 will improve understanding of results to color blind readers. You can see the corrected Figure 1 and 2 in the revised manuscript.

  1. Page 9 paragraph 3.5, the authors talk about subdividing the samples into stage 1,2 and stage 3,4, then talk about higher expression of certain genes being associated with worse overall survival in all stages. This was confusing. The authors should first talk about associations in all stages, then describe how they were categorized as early or late stage and then show results from the two categories.

Answer: Thank you for your kind advice. As you mentioned, we clarified the description as follow: Prior to the Kaplan-Meier survival analysis using the Kaplan-Meier plotter, all patients with serous ovarian cancer were categorized into disease stages as early (including stage I and II) and as advanced (including stage III and IV) stage. Then, we showed the results of association with prognosis regardless of stages in Figure 6. Next, we showed the results of association with prognosis in the early stages (including stage I and II, Figure 7) and the advanced stages (including stage III and IV, Figure 8). You can see the corrected description in the revised manuscript (page 10, line 263).

  1. Figure 6 has writing in a very tiny font that is illegible. The font needs to be increased.

Answer: Thank you for your kind advice. As you mentioned, we adjusted the font size in Figure 6, 7, and 8. You can see the corrected Figure 6, 7, and 8 in the revised manuscript.

  1. The first line of the Discussion says that ovarian cancer is one of the leading causes of death among females. This should be toned down. In female cancer deaths, ovarian cancer ranks fifth.

Answer: Thank you for your kind advice. As you mentioned, we toned down the description. You can see the corrected description in the revised manuscript (page 14, line 300).

  1. Page 12, line 234, the authors state, “Higher CDCA3 and UBE2C gene expression was significantly associated with higher histologic grades, while neither gene was significantly associated with other clinical features.” Please list the other clinical features that were analyzed.

Answer: Thank you for your kind advice. As you mentioned, we corrected the description. You can see the corrected description in the revised manuscript (page 14, line 292).

Reviewer 2 Report

The authors have analysed 3 databases for genes differentially expressed between normal ovarian epithelia and serous ovarian carcinoma. They then use the TCGA data set to create survival plots for these genes, and identify 2 genes (CDC3A and UBEC2) that they believe to be important prognostic markers for serous ovarian cancer.

Suggested improvements:

Many areas need writing, and the article needs proof reading for sense.

Introduction contains too much methodology.

The first 2 paragraphs of the discussion is repetitive of the introduction. I suggest incorporating this into the introduction and starting the discussion at paragraph 3.

The data presented shows that CDC3A and UBEC2 are not significantly associated with worse OS in advanced stages (Fig. 8), so the conclusion is not supported.

Author Response

August 23rd, 2021

Dear Reviewer 2,

Bioinformatic analysis for influential core gene identification and prognostic significance in advanced serous ovarian carcinoma

     We sincerely appreciate the time and effort of you and the referees spent in considering and evaluating our manuscript (medicina-1325979) entitled with “Bioinformatic analysis for influential core gene identification and prognostic significance in advanced serous ovarian carcinoma” for publication in the Medicina. Having received the kind comments by you, we revised our manuscript with attention to each of the comments by you. We appreciate the reviewer 2 very much, who raised the very important critiques to strengthen the claim of our manuscript. We have given very careful consideration to the suggestions and have revised our manuscript. We incorporated new information in the revised version of our manuscript.

     We have fully described the manuscript addressing to the particular issue raised. We have responded all the comments by the reviewer 2 point-by-point as follows. We marked light blue on the corrected or added parts on the manuscript in comparison with un-revised manuscript.

     Thank you very much in advance for your time and effort involved in arranging review process.

Yours Sincerely,

Shin Kim, M. D., Ph. D.

The corresponding author

Reviewer’s comments

# Reviewer 2

The authors have analysed 3 databases for genes differentially expressed between normal ovarian epithelia and serous ovarian carcinoma. They then use the TCGA data set to create survival plots for these genes, and identify 2 genes (CDC3A and UBEC2) that they believe to be important prognostic markers for serous ovarian cancer.

Suggested improvements:

Many areas need writing, and the article needs proof reading for sense.

Introduction contains too much methodology.

Answer: Thank you for your kind advice. We agreed your opinion. In fact, the reason why there are so many methodologies in the Introduction is because it was considered necessary to list the bioinformatics research methods used during this study. Additionally, this is because, although various bioinformatic research methods have been and are still being developed, it is intended to present the justification of the research methods used in this study. Please understand.

The first 2 paragraphs of the discussion is repetitive of the introduction. I suggest incorporating this into the introduction and starting the discussion at paragraph 3.

Answer: Thank you for your kind advice. We agreed your opinion that the first 2 paragraphs of the discussion is repetitive of the introduction. Actually, the first paragraph is written to further emphasize the importance of treating ovarian cancer and discovering prognostic factors. The second paragraph describes in more detail how the DEG analysis, which is the starting point of the various bioinformatics analysis conducted in the presents study, was carried out. Moreover, this paragraph is written to provide more detailed information to researchers who need bioinformatic analysis research. If there is no problem with the length of the discussion section, please understand the first two paragraphs of the description.

The data presented shows that CDC3A and UBEC2 are not significantly associated with worse OS in advanced stages (Fig. 8), so the conclusion is not supported.

Answer: Thank you for your kind advice. As you mentioned, we carefully checked the results of Figure 8. We found that we used the wrong result of Figure 8 in the manuscript. We corrected the Figure 8 and changed the figure. You can see the corrected Figure 8 in the revised manuscript.
